# High-order harmonics measured by the photon statistics of the infrared driving-field exiting the atomic medium

N. Tsatrafyllis[1,2], I.K. Kominis[2,3], I.A. Gonoskov[4] & P. Tzallas[1]

High-order harmonics in the extreme-ultraviolet spectral range, resulting from the strong-field laser–atom interaction, have been used in a broad range of fascinating applications in all states of matter. In the majority of these studies the harmonic generation process is described using semi-classical theories which treat the electromagnetic field of the driving laser pulse classically without taking into account its quantum nature. In addition, for the measurement of the generated harmonics, all the experiments require diagnostics in the extreme-ultraviolet spectral region. Here by treating the driving laser field quantum mechanically we reveal the quantum-optical nature of the high-order harmonic generation process by measuring the photon number distribution of the infrared light exiting the harmonic generation medium. It is found that the high-order harmonics are imprinted in the photon number distribution of the infrared light and can be recorded without the need of a spectrometer in the extreme-ultraviolet.

[1] Foundation for Research and Technology-Hellas, Institute of Electronic Structure & Laser, PO Box 1527, GR-71110 Heraklion, Greece. [2] Department of Physics, University of Crete, 71103 Heraklion, Greece. [3] Institute of Theoretical and Computational Physics, University of Crete, 71103 Heraklion, Greece. [4] Max Planck Institute of Microstructure Physics, Weinberg 2, D-06120 Halle, Germany. Correspondence and requests for materials should be addressed to P.T. (email: ptzallas@iesl.forth.gr).

Strong-field laser-atom interactions induced by intense laser pulses led to the observation of high-order harmonics (HOH) in the extreme-ultraviolet (XUV) spectral range[1,2]. Due to its coherent properties, this radiation has been used in a broad range of applications ranging from ultrafast XUV science[3,4] to high-resolution spectroscopy in the XUV[5,6]. In the majority of these studies the harmonic generation process is described by semi-classical approaches (three-step model)[7], treating the electron quantum mechanically while, due to the high-photon number, the electromagnetic field of the driving laser pulse is treaded classically. Hence, the driving infrared laser radiation is ordinarily understood not to be affected by the interaction. On the experimental level, due to the short wavelength of the generated HOH, measurements are typically performed under high vacuum conditions using specialized XUV equipment.

Another major area of research in optical science, decoupled from ultrafast and strong-field physics, has been quantum optics, founded on the quantized radiation field. The state of the latter is directly affected by the light-matter interactions usually studied with low-photon number light sources. Central to these studies was the measurement and interpretation of light's intensity fluctuations[8,9].

Here we experimentally demonstrate the unification of the above fields, as we obtain the HOH spectrum by measuring the energy distribution (or equivalently the photon number) of the infrared light exiting the harmonic generation medium, that is, without using an XUV spectrometer. The quantum-optical nature of the measurement relies on the fact that we do not solely exhibit energy conservation in the infrared-XUV interaction, which is an expected aggregate effect, but we explicitly measure photon number distribution, which has a quantum-optical nature. This is achieved by utilizing an XUV/infrared correlation approach, to subtract the large background (where the signal of interest is built in) resulting from the initial infrared photon number and interaction processes not leading to XUV emission. It is found that the infrared photon number distribution consists of a series of well-resolved peaks corresponding to the HOH spectrum and containing all its well-known features (plateau and cutoff region, infrared intensity dependence).

## Results

### Theoretical background

The theoretical understanding of the measurement was based on the extension of the three-step semi-classical theory[7] to the quantum-optical regime[10], where the infrared field is treated quantum mechanically. This approach takes into account the back-action of the strong-field laser–atom interaction on the driving infrared laser field (Supplementary Note 1), which is imprinted in the probability distribution of the infrared photons after the interaction with the gas phase medium (Fig. 1). Initially (Fig. 1a) the incident multi-cycle infrared light in a coherent state has Gaussian photon number distribution. After the coherent interaction with $n_a$ atoms towards the generation of XUV radiation (Fig. 1b), the distribution consists of peaks, named infrared (IR)-harmonics (Fig. 1c), which correspond to the HOH order $q = \omega_{XUV}/\omega_{IR}$ (Fig. 1d). Since each atom absorbs $q$ infrared photons towards the generation of the $q$-order harmonic, these peaks represent the missing infrared photon number $N_q^{(IR)} = q n_a$. Note, that $n_a$ is a parameter, which can be obtained by the detected photon number taking into account the propagation effects and phase matching conditions[11,12]. In particular[10], it is found that the infrared photon number probability distribution $P_n$ after the interaction with $n_a$ atoms of the medium depends on the probability amplitudes $A_n$ and a phase $\Phi(t_i, t_r)$ ($t_i, t_r$ are the ionization and recombination times, respectively) (Supplementary Note 1). Since the amplitudes $A_n$ depend on

$N_q^{(IR)}$ and $t_i, t_r$ are directly linked to the XUV spectral phase distribution, measuring infrared-harmonics can reveal the XUV spectrum and in principle the spectral phase distribution of the XUV light. Here we demonstrate the former.

### Experimental approach and results

The experimental set-up is shown in Fig. 2a. A 25 fs Ti:Sapphire laser pulse of 800 nm carrier wavelength, 10 Hz repetition rate and $\approx 0.6$ mJ energy per pulse was split in two by a beam splitter. The transmitted beam (IR$_0$), with photon number $\approx 10^{15}$ photons per pulse, was focused by a 30 cm focal length lens (L) onto a Xenon gas-jet where the HOH were generated. After passing through the HOH generation region, the infrared beam was attenuated by a factor of $T^{-1} = 3 \times 10^6$ ($T$ is the transmission coefficient) and the exiting beam (IR$_1$) was recorded by the photodiode PD1. The beam reflected from the beam splitter, after attenuation, was recorded by the photodiode PD2 and the generated XUV radiation by a photo-multiplier (PMT). The signals of PD1, PD2 and PMT, denoted by $S_{PD1}$, $S_{PD2}$ and $S_{PMT}$, respectively, were simultaneously recorded for each laser shot. $S_{PD1}$ was used for recording the probability distribution of the energy of IR$_1$, while $S_{PD2}$ and $S_{PMT}$ were used for removing the energy fluctuations of the laser and the unwanted background caused by processes irrelevant to the XUV emission, respectively. The background is an XUV-infrared non-correlated signal which introduces an offset and reduces the visibility of the infrared distribution which is correlated with the XUV emission. When the Xenon gas-jet was off, $S_{PD1}$ and $S_{PD2}$ were balanced (Fig. 2b) with corresponding photon number $N_0 \approx 3.3 \times 10^8$ photons per pulse. When the gas-jet was on, $S_{PMT}$ increased while $S_{PD1}$ was reduced (Fig. 2b).

This observation, which is in fair agreement with the predictions of the ADK theory[13] (reduction by a factor of $\sim 1.5$ that is, $N_0' = N_0 - N_{abs}^{(IR)} - N_q^{(IR)} \approx 2 \times 10^8$ photons per pulse), reflects energy conservation, where $N_q^{(IR)}$ and $N_{abs}^{(IR)}$ photons of the infrared beam are absorbed towards HOH generation and all other processes, respectively. $N_0'$ reflects the remained infrared photon number resulted after the absorption of $N_q^{(IR)}$ and $N_{abs}^{(IR)}$ photons. Moreover, both the reduction of $S_{PD1}$ (increase of $S_{PD2} - S_{PD1}$) and the enhancement of $S_{PMT}$ have the same non-linear dependence with IR$_0$ intensity (Fig. 2c), also in agreement with the ADK theory.

The calibration of the PMT signal to the photon number was done by taking into account the quantum efficiency of the detector, the reflectivity of the XUV optics and the XUV filter transmission (Methods section). For the infrared beams, the signal of the diodes was calibrated to the photon number using three different approaches which lead to the same result: (I) by corresponding the diode signal to the infrared energy value (measured by a power-meter) taking into account the transmission of the optical elements and the neutral density filters (with transmission coefficient $T$, with $T^{-1} = 3 \times 10^6$), which have been used in order to avoid saturation effects in the diode; (II) using the specifications (responsivity and load resistance) of the photodiodes; and (III) by means of a single-photon counter.

Although an accurate calculation of the probability distribution requires the consideration of the infrared laser bandwidth and the propagation effects in the medium, a rough estimation that can provide an indicative value of the infrared photons absorbed towards the harmonic emission and the main features of their distribution can be given by correcting the measured harmonic photon number for the XUV absorption effects. The XUV photon number at the output of the Xenon gas found to be in the order of $N^{(XUV)} \sim 10^8$ photons per pulse translating to $N_q^{(XUV)} \approx (N^{(XUV)}/5) \approx 2 \times 10^7$ photons per harmonic (where 5 is the

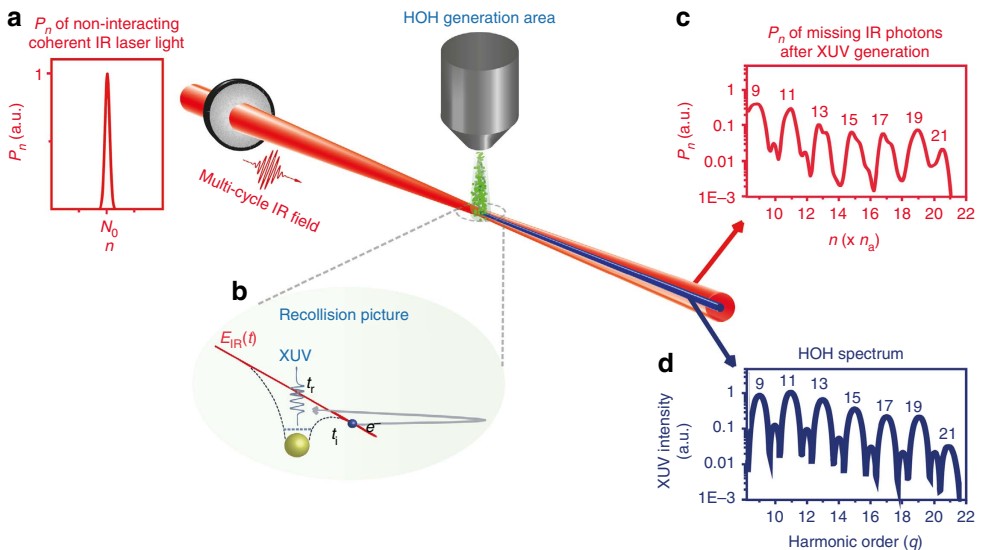

**Figure 1 | Influence of the HOH generation process on the driving laser photon statistics.** (a) Probability distribution, $P_n \propto \exp\left[-\frac{(n-N_0)^2}{2N_0}\right]$, of the non-interacting multi-cycle coherent infrared-light photon number $n$, where $N_0 \gg 1$ is the initial photon number. (b) HOH generation area and recollision picture. The electron tunnels through the atomic potential at $t_i$, accelerates in the continuum under the influence of the laser field and emits XUV radiation at the recombination time $t_r$. (c) Probability distribution of the absorbed infrared photons after the HOH generation area calculated using the full quantum mechanical approach for $n_a = 500$ atoms. (d) HOH spectrum calculated using the semi-classical model. The calculations have been done for Xenon gas and laser intensity $8 \times 10^{13}$ W cm$^{-2}$.

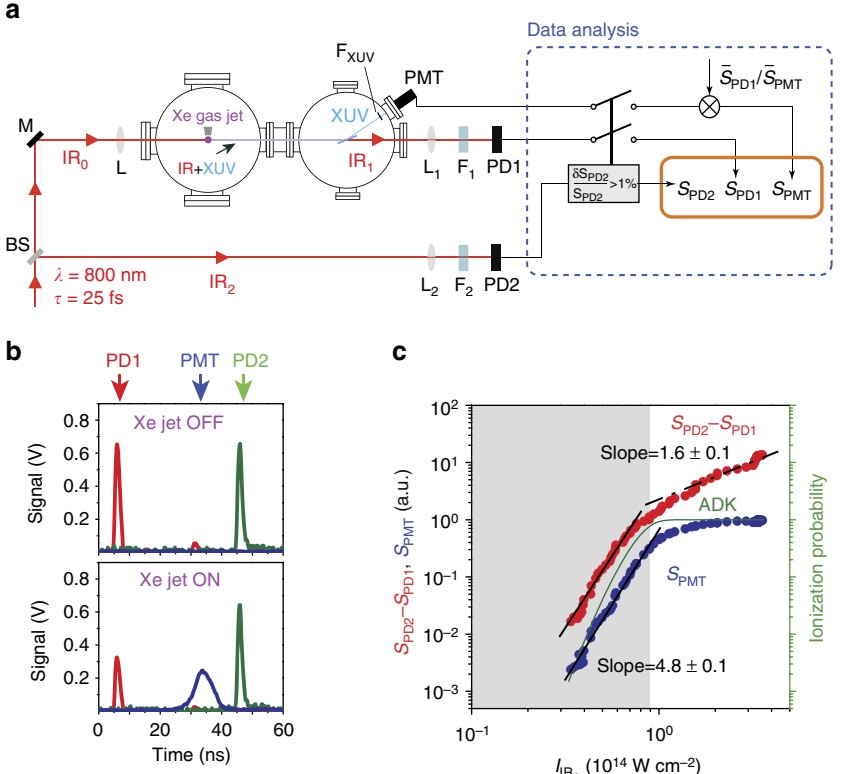

**Figure 2 | Method used for recording the infrared energy distribution exiting the gas medium.** (a) Experimental set-up and data analysis. BS, infrared beam splitter; M, infrared plane mirror; L,L$_1$,L$_2$, infrared focusing lenses; PMT, XUV photomultiplier; IR$_0$ and IR$_1$, incoming and outing infrared beams; F$_{1,2}$ and F$_{XUV}$, infrared and XUV neutral density filters; PD1, PD2, infrared photodiodes. The relative fluctuations of $S_{PD2}$ are denoted by $\delta S_{PD2}/S_{PD2}$ and are used to gate the data. (b) $S_{PD1}$, $S_{PD2}$ and $S_{PMT}$ signals recorded when the Xenon gas-jet is OFF and ON. (c) Dependence of the $S_{PD2}$–$S_{PD1}$ (red-dots) and $S_{PMT}$ (blue-dots) on infrared intensity for Xenon with 100 shots accumulated for each point. The shaded area shows the intensity range below ionization saturation. The green solid-line depicts the calculated (ADK theory) ionization probability. The slope of $\approx 1.6$ observed for $S_{PD2}$–$S_{PD1}$ at intensities $>10^{14}$ W cm$^{-2}$ is attributed to volume effects.

number of harmonics lying in the plateau region). Taking into account the experimental conditions, it turns out that the XUV photon number is reduced due to absorption effects in the medium by a factor of $A \approx 5 \times 10^4$ (Methods section). Considering that $q$ infrared photons are required for the generation of the $q$th harmonic, the number of the infrared photons absorbed towards harmonic emission is $\tilde{N}_q^{(IR)} = AqN_q^{(XUV)} \approx 2 \times 10^{13}$ photons per pulse (for $q = 17$), translating to $N_q^{(IR)} = \tilde{N}_q^{(IR)}T \sim 10^7$ photons per pulse at PD1. In addition, the photon number difference between consecutive peaks ($\Delta q = 2$) in the IR$_1$ probability distribution is expected to be $\Delta N_q^{(IR)} = \widetilde{\Delta N_q^{(IR)}}T \sim 10^6$ photons per pulse (where $\widetilde{\Delta N_q^{(IR)}} \approx A \cdot \Delta q \cdot N_q^{(XUV)} \approx 2 \times 10^{12}$ photons per pulse). Although the above estimations are rough, they depict the feasibility of performing infrared photon distribution measurements for revealing the high-order harmonic spectrum using conventional detection techniques. In addition, we would like to point out that, although the interpretation of our measurement is independent of the value of $n_a$, the derivation of $n_a$ from the detected number of XUV/infrared photons requires careful consideration of the experimental conditions and propagation effects of the XUV/infrared fields in the harmonic generation medium[11,12], which is beyond the scope of this work. Taking into account the above, $S_{PD1}$ is expected to have a distribution located around $N_0' \approx 2 \times 10^8$ photons per pulse with the harmonic peak structure being about 2 orders of magnitude smaller.

The probability distribution of the energy of IR$_1$ reaching PD1 is shown in Fig. 3a. To minimize the effect of the laser's energy fluctuations, we gate $S_{PD2}$ and keep only those pulses having energy stability at the level of 1%. With the Xenon jet off, $S_{PD1}$ has the same amplitude as $S_{PD2}$ (Fig. 2b) and similar fluctuations. With the Xenon jet on, the distribution of $S_{PD1}$ is shifted towards lower intensities, $2 \times 10^8$ photons per pulse, and broadens significantly (Fig. 3a). The broadening results from the strong-field laser-atom interaction. A peak structure associated with the generation of harmonics is not clearly visible in this distribution. To enhance the peak visibility, we take advantage of the dependence shown in Fig. 2c and the measured anti-correlation of $S_{PD1}$ with $S_{PMT}$. Before doing so, we normalize $S_{PMT}$ (see the data analysis part of Fig. 2a) so that it has the same average value as $S_{PD1}$, that is, we multiply $S_{PMT}$ with the ratio $\bar{S}_{PD1}/\bar{S}_{PMT}$. The widths of the two distributions are the same, and their correlation map is shown in Fig. 3b. Since the centre of this distribution moves in an anti-correlated way when changing the infrared intensity (Fig. 2c), we define a cut in the distribution of Fig. 3b along the anti-correlated diagonal (Methods section), resulting in the distribution of Fig. 3c.

The infrared-harmonic distribution reveals the HOH peaks, as can be validated by the following systematic checks. First, the distribution exhibits all the known features (plateau and cutoff regions) of an HOH spectrum recorded by a conventional XUV spectrometer. It contains a series of well-resolved peaks, which correspond to the 9th–23rd harmonic orders. As expected for

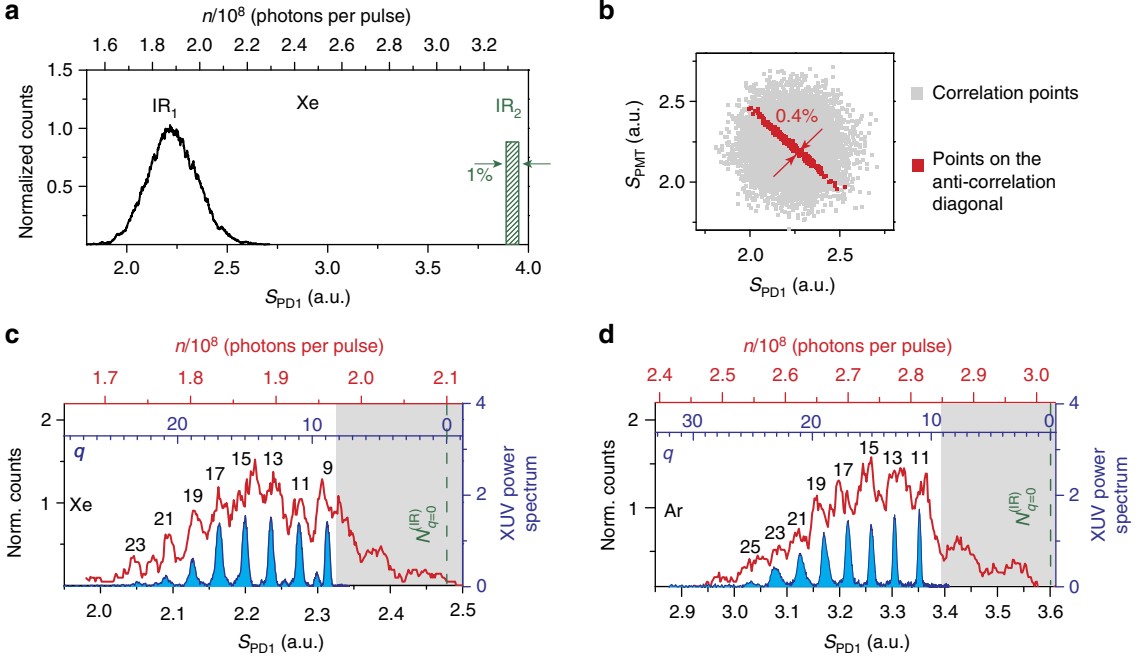

**Figure 3 | Photon statistics of the infrared field exiting the HOH generation medium. (a)** The black-line shows the $S_{PD1}$ distribution after Xenon gas and after gating PD2. The green-line-shaded area shows the gated $S_{PD2}$. **(b)** Correlation of $S_{PMT}$ with $S_{PD1}$ after the PD2 gating (grey points). Cutting on the anti-correlation diagonal (red points) leads to **(c)** Final infrared-harmonic distribution after Xenon gas. The distribution is recorded for infrared intensities $\approx 8 \times 10^{13}$ W cm$^{-2}$. The value of $\Delta N_q^{(IR)}$ is found to be $\approx 3.4 \times 10^6$ photons per pulse. **(d)** Final infrared-harmonic distribution after Argon gas. The value of $\Delta N_q^{(IR)}$ is found to be $\approx 4.3 \times 10^6$ photons per pulse. The distribution is recorded for infrared intensities slightly higher than Xenon case that is, $\approx 10^{14}$ W cm$^{-2}$, keeping the initial infrared energy very close to the Xenon case that is, $N_0 \approx 3.3 \times 10^8$ photons per pulse. The blue shaded areas (and the corresponding blue axes) show the harmonic spectrum measured with the XUV spectrometer. The upper $x$ axis of **a,c,d** noted with $n$ shows the measured photon number, while $N_q^{(IR)}$ is the absorbed photon number, which increases with $q$. The grey shaded areas in **c,d** correspond to the harmonics laying below the ionization potential (IP) of Xenon and Argon, respectively. We note that, as in the majority of the HOH generation experiments[12], in both gases (Xe, Ar) the intensity of the laser in the HOH generation regime was kept just below the ionization saturation intensity that is, $\leq 10^{14}$ W cm$^{-2}$ where the value of the Keldysh parameter is $\gamma = (IP/2U_p)^{1/2} \approx 1$ ($U_p$ is the ponderomotive energy of the electron). This value is typical for the majority of the HOH generation experiments where the ionization of an atom falls in the tunnelling regime.

HOH generation in Xenon[12], the 9th–15th infrared-harmonics are laying in the plateau spectral region while the infrared-harmonics $> 17$th belong in the cutoff spectral region. Second, we compare the $S_{PD1}$ distribution with a HOH spectrum recorded using a conventional XUV spectrometer. It is found that the relative intensity of the infrared-harmonics matches the intensity of the harmonics measured with the XUV spectrometer (blue shaded area in Fig. 3c). Third, the distribution was also recorded with Argon gas as harmonic generation medium. The 11th–19th infrared-harmonics are in the plateau spectral region, while the infrared-harmonics $> 21$st are in the cutoff spectral region. Again, infrared-harmonics intensity is consistent with the spectrum measured with the XUV spectrometer (blue shaded area in Fig. 3d). The green dashed line (noted as $N_{q=0}^{(IR)}$) is positioned at the number of measured infrared photons $n$ corresponding to $N_{q=0}^{(IR)}$, which is obtained by extrapolating to $q = 0$ the linear dependence of $N_q^{(IR)}$ on $q$ (Supplementary Note 2). This value of $n$ coincides with the value corresponding to $q = 0$ obtained from the harmonic spectrum (blue $x$ axis in Fig. 3c,d) and reflects the remained infrared photon number resulting from the absorption due to processes other than harmonic emission $\left(N_{abs}^{(IR)}\right)$. At the green dashed line, $N_{q=0}^{(IR)} = 0$ and the measured photon number (upper red $x$ axis) is $N_0 - N_{abs}^{(IR)}$. Due to the weaker ionization of Argon the position of $N_{q=0}^{(IR)}$ (where $N_0 - N_{abs}^{(IR)} \approx 3 \times 10^8$ photons per pulse) is closer to $N_0$ compared to the Xenon case (where $N_0 - N_{abs}^{(IR)} \approx 2.1 \times 10^8$ photons per pulse), while as expected for HOH generation in Argon the number of infrared-harmonic peaks was increased and the cutoff position moved to higher harmonics. Fourth, we observe the shift of the cutoff region to lower order infrared-harmonics when reducing the intensity of the driving infrared laser (Fig. 4). This measurement is found to be in fair agreement with the results obtained (black-dashed lines) using the full quantum mechanical theoretical approach[10]. In addition to the above checks, the harmonic peak structure observed in the XUV distributions recorded by the PMT (Supplementary Note 3), and corresponding to Fig. 3c,d, further supports the above findings. We finally note, that understanding the rapid reduction of the infrared-harmonics laying below the ionization potential of Xenon and Argon (grey shaded area in Figs 3c,d and 4) requires more elaborate consideration of perturbative effects and ionization phenomena[14], which is beyond the scope of this work.

## Discussion

Concluding, we experimentally demonstrated that the XUV harmonic spectrum generated in gas phase media is imprinted in the probability distribution of the energy of the infrared light exiting the harmonic generation medium. These findings constitute an experimental demonstration of the quantum-optical nature of the high-order harmonic generation process and pave the way for quantum-optical studies in the strong-field region. In particular, they allow to perform experiments in ultrafast XUV science using conventional infrared diagnostics.

## Methods

**Experimental procedure.** A 25 fs infrared laser pulse was split in two by a beam splitter. The transmitted infrared beam ($IR_0$), after passing through the interaction region, was transmitted through optical elements and neutral density filters ($F_1$) having overall transmission coefficient $T^{-1} = 3 \times 10^6$. The output infrared laser beam ($IR_1$) was recorded by the PD1. The number of infrared photons reaching PD1 when the gas-jet was switched-off was $N_0 \approx 3.3 \times 10^8$ photons per pulse. The beam reflected from the beam splitter was recorded by PD2. When the gas-jet was off, the signals of PD1 and PD2 were balanced by the neutral density filter $F_2$. The piezo-based pulsed nozzle of $\approx 0.8$ mm orifice diameter, which was constricted according to the configuration of ref. 15, provides a gas density in the interaction region $\rho \approx 3 \times 10^{18}$ atoms per cm³. The beam was focused $\approx 0.5$ mm below the nozzle orifice resulting to a medium length $L_{med} \approx 1.2 \pm 0.1$ mm. In this

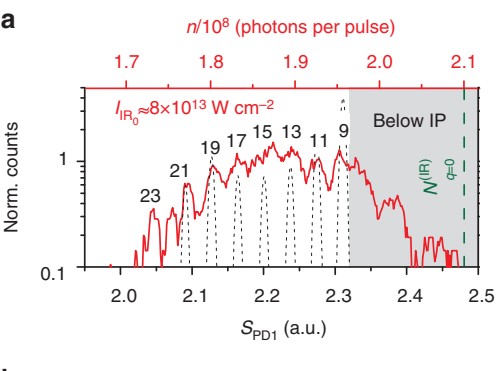

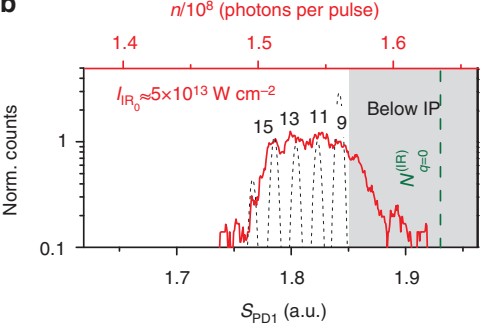

**Figure 4 | Dependence of infrared-harmonic distribution on the intensity of the infrared beam.** Infrared-harmonic photon distribution (red-line) recorded using Xenon gas for two infrared intensities ($I_{IR_0}$). (**a**) Log scale of the infrared distribution of Fig. 3c, where $I_{IR_0} \approx 8 \times 10^{13}$ W cm$^{-2}$ ($N_0 \approx 3.3 \times 10^8$ photons per pulse). (**b**) $I_{IR_0} \approx 5 \times 10^{13}$ W cm$^{-2}$ ($N_0 \approx 2 \times 10^8$ photons per pulse). In this case the values of $\Delta N_q^{(IR)}$, and the position corresponding to $N_{q=0}^{(IR)}$ (green dashed line) in the distribution are found to be $\approx 1.7 \times 10^6$ photons per pulse and $\approx 1.6 \times 10^8$ photons per pulse, respectively. In both graphs, the black-dashed lines show the theoretical infrared-harmonic distribution calculated for $10^8$ photons per pulse, taking into account the resolution of the measurement. The upper red $x$ axis noted with $n$ shows the measured photon number, while $N_q^{(IR)}$ is the absorbed photon number, which increases with $q$.

configuration the confocal parameter is more than an order of magnitude longer than the length of the medium and thus the intensity of the laser beam along the propagation in Xenon gas can be considered constant. The intensity of the laser in the Xenon gas-jet was kept below $10^{14}$ W cm$^{-2}$, the Xenon ionization saturation intensity. The produced XUV light was reflected into a PMT by a multilayer-infrared-antireflection coating plane mirror placed at 75°, after passing through an Aluminium filter of 150 nm thickness. The photon number of the XUV radiation just after the Xenon gas was $N^{(XUV)} \sim 10^8$ photons per pulse that is, $N_q^{(XUV)} \approx (N^{(XUV)}/5) \sim 2 \times 10^7$ photons per harmonic (where 5 is the number of harmonics lying in the plateau region). This value was obtained by taking into account the relative harmonic amplitudes measured by the XUV spectrometer (Fig. 3c), the harmonic signal measured by the PMT, the quantum efficiency ($\sim 10\%$) and the gain ($10^2$–$10^3$) of the PMT which is used in the non-saturated region, the transmission of the Aluminium filter ($\sim 70\%$) and the reflectivity of the plane mirror ($\sim 50\%$). The HOH spectrum shown in blue shaded area in Fig. 3c was recorded using a conventional XUV spectrometer (without the Aluminium filter) and the same experimental conditions as those used for the measurement of the $IR_1$ intensity distribution. We note, that mainly due to the uncertainty of the PMT gain, the value of $N^{(XUV)}$ is obtained with an accuracy of one order of magnitude.

**Correlation map of Figure 3b.** Figure 3b depicts the joint distribution of the XUV and transmitted infrared intensity. The widths of the individual distributions are seen to be the same (relative width about 7%). This is because both intensities have the same dependence on the incident infrared laser power, and the majority of the points on the joint distribution results from XUV/infrared uncorrelated events mainly associated with $N_{abs}^{(IR)}$. However, we know that the centre of the joint distribution of Fig. 3b moves in an anti-correlated way with varying infrared intensity, as shown in Fig. 2c. Furthermore, we know the position of this centre with precision much larger than the widths of the individual distribution. In particular, the data consists of about 10,000 points, hence the position of the centre is known to within $7\% \times 10,000^{-1/2} \approx 0.1\%$. To enhance the infrared-XUV

correlation and produce the infrared-harmonic peaks, we perform a cut along the anti-correlated diagonal of Fig. 3b with width 0.4%. This is chosen to be larger than the precision of 0.1%, because it practically leads to an optimum visibility and statistical significance of the infrared-harmonic peaks.

**Calibration of the infrared-harmonic spectrum.** The identification of the harmonic spectrum on the infrared-harmonic distribution was done using the linear dependence of $N_q^{(IR)}$ on $q$, setting the cutoff harmonics of the spectra recorded by the XUV spectrometer at the cutoff values of the infrared-harmonic distribution. In this way, the number of the harmonic orders, the constant spacing and the relative harmonic amplitudes recorded by the XUV spectrometer matches the peaks of the infrared-harmonic distribution. Also, this calibration provides a value for the position of $N_{q=0}^{(IR)}$ in the distribution which coincides with the value of $q = 0$ obtained by the harmonic spectrum (blue $x$ axes in Fig. 3c,d; Supplementary Note 2). In addition, the infrared photon number which corresponds to the ionization potential of the atoms matches the region where the infrared distribution drops rapidly (grey shaded areas in Figs 3 and 4).

**Estimation of the infrared photon number.** To roughly estimate the main features of the infrared photon distribution correlated with the harmonic emission, we correct the measured harmonic photon number $N^{(XUV)}$ for the XUV absorption effects (single-XUV-photon ionization) in the Xenon gas[12]. For our experimental conditions where $L_{med} \approx 1.2$ mm, the XUV absorption length caused by single-XUV-photon-ionization process is $L_{abs}^{(XUV)} = 1/\rho\sigma^{(1)} \approx 100$ μm (cross section of Xenon $\sigma^{(1)} \approx 3 \times 10^{-17}$ cm$^2$). As $L_{med} >> L_{abs}^{(XUV)}$ and $L_{coh} >> L_{abs}^{(XUV)}$ (where $L_{coh} = \pi/\Delta k$ is the coherent length[16,17], $\Delta k = k_L - q k_L$ and $k_L$ is the wave number of the fundamental) it follows that the present experiment is conducted in the XUV absorption saturation regime. In this context the XUV photon number reaches the value of $N^{(XUV)} \sim 10^8$ photons per pulse at the beginning of the medium and remains constant along the propagation as the XUV losses induced by the single-photon ionization are getting balanced by the infrared absorption which takes place along the propagation in the whole medium. Considering the medium as a single absorbing filter having exponential dependence on the medium length it follows that XUV photons are reduced due to absorption over the whole medium length (integration over the medium length) by a factor of $A \approx 5 \times 10^4$. Considering that $q$ infrared photons are required for the generation of the $q$th harmonic, the number of the infrared photons absorbed towards harmonic emission is $\tilde{N}_q^{(IR)} = A q N_q^{(XUV)} \approx 2 \times 10^{13}$ photons per pulse (for $q = 17$).

**Data availability.** The data that support the findings of this study are available from the corresponding author upon request.

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

## Acknowledgements
We acknowledge support by the Greek funding program NSRF (ERC-07) and the European Union's Seventh Framework Program FP7-REGPOT-2012-2013-1 under grant agreement 316165.

## Author contributions
N.T. performed the experiment, the theoretical calculations and contributed to the data analysis; I.K.K. contributed on the manuscript preparation and data analysis; I.A.G. contributed on the manuscript preparation and data analysis; P.T. conceived the idea and contributed in all aspects of the present work.

## Additional information

**Competing interests:** The authors declare no competing financial interests.

