## [Peer Review File · Nature Communications]

Reviewers' comments:

Reviewer #1 (Remarks to the Author):

In this paper the authors analyze photon number distribution of infrared (IR) laser exiting the medium in the process of high-order harmonics generation (HHG). They relate the photon loss with the HHG spectrum and propose this method as an alternative to the standard method of direct measurement of HHG photons in the XUV region. I remark that theoretical analysis of this proposed method has been reported quite recently by the authors, see their Ref. [10]. Below are my specific comments.

1) The authors stated that "the work paves the way for a broader synthesis of strong-field physics and quantum optics" without any further elaboration in the manuscript about the potential applications of this method. More importantly, it is quite clear that the current manuscript simply provides an alternative method to measure HHG spectrum, and nothing more – assuming that the method is correct.

2) There is no doubt that HHG should have some imprints on the distribution of IR photons. The question is how we decode this information. The authors' method is not quite convincing to me. The main reason is that the anti-correlated signals [between PD1 and PMT signals -- see their Fig. 3(b)] might still be contaminated due to the excitation of the bound electrons to ATI electrons which correspond to the same number of photon absorption as the HHG process at any given harmonic. In fact, an electron that absorbs q photons might or might not recombine with the parent ion. My understanding is that the current experiment is still not a coincidence experiment so it can't exclude those possibilities in an ensemble of a huge number of atoms. To partially clarify this issue I think the authors should provide a figure for PMT distribution to show what they actually measured.

3) There are few other technical details that are not clear to me. For example, the authors' estimate of IR photon number and the XUV photon number reduction is based entirely on XUV photon absorption. In reality, we know that HHG process depends sensitively on the macroscopic phase-matching condition. It is not quite clear if their method is able to detect this phase-matching effect in IR photon distribution. Another example is related to the contamination mentioned in (2). In fact, it is not quite clear to me if the method is capable of detecting, say, the Cooper minimum in HHG spectra from Ar.

In conclusions, the current manuscript presents some interesting results and therefore it definitely deserves to be published in some form. However, based on the above arguments I do not think it qualifies for Nature Communications.

Reviewer #2 (Remarks to the Author):

Article ID: NCOMMS-16-21519-T

Title: High order harmonics measured by counting the photons of the infrared driving laser pulse

The manuscript "High order harmonics measured by counting the photons of the infrared driving laser pulse" by N. Tsatrafyllis and coworkers presents the experimental evidence of a new investigation method in attosecond science. The approach the authors present in their manuscript is very intriguing and will possibly lead to new interesting developments. For this reason I think that this work deserves publication in Nature Communications. Nevertheless, I cannot recommend publication as it is for the following reasons:

(1) I found the claims in the abstract and in the conclusions a pretentious. The authors write: "Here,

we reveal the quantum-optical nature of the HOH generation process by recording the HOH spectrum without the need of XUV spectrometer." I do not see how recording the HOH spectrum without the spectrometer automatically implies that one reveals the quantum-optical nature of the HOH. The authors should explain better what they mean with quantum-optical nature. After all, a harmonics spectrum is composed by peaks at an energy corresponding to precise multiples of the fundamental frequency. By energy conservation, this implies that every harmonic has taken a finite (odd) number of IR photons. What the authors show is the corresponding effect on the IR generating pulse. The two aspects seem to me to be linked since they represent two distinct ways to look at the same process. So it is not clear to me why one should not be able to conclude the quantum-optical nature of HHG already from the spectral distribution of the harmonics.

On line 133, the sentence "These findings constitute an experimental demonstration of the quantum-optical nature of the strong-field light-atom interaction" pretends to be more general than what it should be. After all strong-field light-atom interaction describes a multitude of effects (e.g. above threshold ionization, tunneling ionization, HHG, ...) and the authors just investigated one of them.

One line later the authors claim: "In particular, they allow the development of new high-photon-number non-classical light sources". This theme is never addressed in the rest of the manuscript and I do not see how counting the IR photons which are anti-correlated to the XUV photons will help to develop a new non-classical light source. The authors should justify better their claim or remove it.

(2) I found the description of the origin of the IR photon number at page 3 and 4 badly written and confusing. There are quantities that are not properly introduced. Let me be more precise. At line 59 the authors write: "it is found that the IR photon number probability distribution after the medium depends on the probability amplitudes A_n and a phase Φ ...". These two quantities (A_n and Φ) neither are introduced explicitly in the manuscript nor in the supplementary. So, without reading Ref. [10], this sentence conveys no message if not that the probability depends on its amplitude and its phase. This is obvious and true for any quantity.

The quantity N_0' at line 77 has not been defined. One can guess that it is the photon number when the HHG has been turned on, but this should be clearly stated.

Lines 82 to 85 are difficult to follow. At this point of the manuscript the reader is left with the impression that a voltage signal on the photodiode magically transforms in photon counts. How this happens is totally unclear. Are the photodiodes calibrated or not? How is the XUV photon number (~ 108 photon/pulse) measured? How one can estimate from that number that we have $N_q(\text{IR})$ of 2×10^{13} ? And what is the quantity T ? It is true that the authors refer to the method section where these quantities are explained, but this part is crucial to understand the whole work and thus should be described better already in the manuscript. In particular because if I take the quantities as they are written in the manuscript I get to an unphysical result. If I take $\Delta N_q(\text{IR}) \sim 107$ (line 84), $N_q(\text{IR}) \sim 2 \times 10^{13}$ (line 83) and the definition of R at line 88, I find that, in order to have $R=2$, q should be 4×10^7 which does not make sense. Can the problem be that $\Delta N_q(\text{IR})$ at line 84 is not the real one, but just the one transmitted through the filter?

I have also found another apparent inconsistency in the method section. At line 193 $N_q(\text{IR}) \sim 2 \times 10^{13}$ is associated to an expected difference of $\Delta N_q(\text{IR}) = 4 \times 10^{12}$. But if I take the definition of $N_q(\text{IR}) = A_q N_q(\text{XUV})$ I should expect that $\Delta N_q(\text{IR}) = 2 A_q N_q(\text{XUV})$. If I put in the values for A and $N_q(\text{XUV})$ I get $\Delta N_q(\text{IR}) = 5 \times 10^4 \times 2 \times 10^7 \times 2 = 2 \times 10^{12}$ which is half of what reported at line 195.

Finally, I find confusing the definition of $N_q(\text{XUV})$. In figure 3 the authors clearly show that the harmonic intensity is not uniform. So I do not expect $N_q(\text{XUV})$ to be constant, but to vary significantly with q . To what does the value 2×10^7 correspond? Is it an average or the value for the harmonics in the plateau?

(3) At line 120 the authors write: "'IR-harmonics" intensity is consistent with the spectrum measured with the XUV spectrometer (blue dots in Fig. 3d).", but in figure 3 I do not see the full XUV spectrum. For the sake of a better comparison and comprehension, the authors should show the full UXV

spectrum instead of the blue dots in fig. 3c and 3d.

(4) I couldn't find any information on the laser repetition rate.

(5) Is the part on the calibration of the IR-harmonic spectrum in the methods implying that one anyway needs the XUV spectrum to calibrate the signal? If so, this should be openly stated in the manuscript too.

Minor comments:

- line 92, Fig 3A is marked with capital letter while all the others are labeled with small letter.
- In fig. 4 the dots are difficult to read. I suggest to make the lines dashed instead of dotted.

Reviewer #3 (Remarks to the Author):

The authors measure the probability distribution of the energy of an ultrashort laser pulse after it has undergone intense nonlinear propagation in a gas, producing high-order harmonics. The probability distribution, conditioned on simultaneous measurements of the energy of the harmonics, appears similar to the spectrum of the harmonics measured on a conventional XUV spectrometer. The authors' interpretation is that this is due to the quantum optical nature of the interaction, which quantizes the number of photons absorbed per gas atom.

I congratulate the authors on their attempt at tackling an interesting unsolved problem – the quantum optical nature of HHG – in a creative and original way. However I have some significant concerns with the concepts and interpretation, as well as the details of the data processing which are crucial links in the authors chain of reasoning. For this reason I do not support publication of the manuscript in its present form, but would re-evaluate my opinion if significant improvements were made which addressed these problems.

The manuscript relies heavily on the concept of the number of atoms with which the light interacts coherently, n_a . This needs more explanation – as presented, it has some serious contradictions with existing knowledge.

1. The authors state that each atom absorbs q IR photons towards the generation of the q -th order harmonic, and as such the number of missing IR photons is $N_q^{(IR)} = qn_a$. This contradicts well established theory (see e.g. Eberly et al. PRA 1992 45 p4706) that the intensity of light forward scattered coherently by a collection of atoms scales as the number of atoms squared. This fact is widely known and built into all macroscopic models of HHG, and is experimentally well established. In the authors' picture, if I have a system with $N_q^{(IR)}$ photons contributing to the q th harmonic, and then I double the number of atoms, then I now have $2N_q^{(IR)}$ photons contributing. The standard viewpoint says I will have 4 times the intensity and hence four times the number of XUV photons. Where do the extra photons come from?

2. I suspect, but am not certain, that the resolution of these conflicting views lies in the fact that the experiment is conducted in the absorption saturated regime. The authors state that the absorption length is much smaller than the target thickness. They do not discuss the coherence length, but it is implied to be much longer. In the absorption saturated regime (see e.g. Constant et al. PRL 82 p1668 1999), the number of harmonic photons per unit cross sectional area is independent of the atomic density, but if the area is scaled then the total number of harmonic photons could scale as the number of atoms. (This is just one possibility...)

3. If we accept the authors statement that each atom absorbs q photons to make one XUV harmonic, then n_a must be harmonic order dependent, or else the harmonic spectrum would be flat. What then does n_a really mean – the number of atoms in a volume of sufficient laser intensity to produce a given harmonic? This idea should be more carefully introduced.

In addition to these conceptual concerns, I found some of the data processing hard to follow, with important details missing.

(1) I couldn't precisely follow all steps in the derivation of R . The authors state that the spacing of $\Delta N_q^{(IR)}$ is proportional to $2Tn_a$. If we assume equality rather than proportionality was intended, then the definition of R follows from substituting $N_q^{(IR)} = qn_a$. This should be made clear.

(2) The procedure for obtaining $N_q^{(IR)}$ is described in the last subsection of the Methods section. In the description, it is not clear which values are resolved by harmonic, and which are totals. It seems that $N^{(XUV)}$ is the total detected harmonic photon number, but the gas absorption used to calculate the emission from the gas must have some dependence on frequency. Then, the quantity $N_q^{(XUV)}$ appears without introduction. Is this the XUV photon number resolved by harmonic? Do the authors use the measured spectrum to split the total number of XUV photons into individual harmonics?

(3) It is stated that $N_{\{q=0\}}^{(IR)}$ is inferred from extrapolating the linear dependence of $N_q^{(IR)}$ on q . I didn't understand this. Partly the problem is notation – is $N_q^{(IR)}$ and experimental observable, in which case the formula $N_q^{(IR)} = AqN_q^{(XUV)}$ is incorrect and needs an additional term? Or is this formula correct, in which case $N_{(q=0)}^{(IR)}$ is identically zero.

(4) The method of estimating the number of XUV photons isn't described in sufficient detail. The authors speak of the XUV photons being reduced by absorption along the medium length. How is this modelled? Do they treat the harmonics as being generated without absorption, and then after generation passing through the gas, modelled as a single filter? Or do they model the simultaneous buildup and absorption of harmonics, which (provided there is phase matching) reaches an equilibrium after several absorption lengths (see e.g. Constant et al. PRL 1999 82). In this case, the number of photons predicted without absorption would be very large, as coherent build-up of the field would produce a signal proportional to the number of atoms squared. This process is crucial, since it is used in evaluating R .

(5) The superimposed XUV spectra seem to agree nicely. With the caveats described above, I was able to understand how the authors scaled the measure photon number to give harmonic order. What about the offset? Apart from maximizing the similarity of the XUV spectra and photon number distribution, is there any justification for assigning the peak of the photon number distribution to a certain harmonic (e.g. harmonic 15 in Fig. 3c).

(6) Are all the arbitrary units the same throughout for S_{PD1} , S_{PD2} , S_{PMT} i.e. can numbers on the vertical axis of Fig. 2© be compared directly to those on the vertical axis of Fig. 3(b) (and likewise for all other plots of the $S_{_}$ signals)?

(7) How were the photodiodes, particularly PD1, calibrated? This is crucial information.

Reply to referees for the manuscript (Nr.: NCOMMS-16-21519-T) with title "*High order harmonics measured by counting the photons of the infrared driving laser pulse*" submitted to Nature Comm. by N. Tsatrafyllis et al.

Our reply to the referees comments are written in blue and italic.

Reply to Referee#1

We thank the referee for the report and his/her recognition that our work is interesting and deserves to be published. The manuscript has been modified taking into account all his/her comments and suggestions.

In this paper the authors analyze photon number distribution of infrared (IR) laser exiting the medium in the process of high-order harmonics generation (HHG). They relate the photon loss with the HHG spectrum and propose this method as an alternative to the standard method of direct measurement of HHG photons in the XUV region. I remark that theoretical analysis of this proposed method has been reported quite recently by the authors, see their Ref. [10]. Below are my specific comments.

Comment #1) The authors stated that “the work paves the way for a broader synthesis of strong-field physics and quantum optics” without any further elaboration in the manuscript about the potential applications of this method. More importantly, it is quite clear that the current manuscript simply provides an alternative method to measure HHG spectrum, and nothing more – assuming that the method is correct.

We agree with the referee. The sentence "the work paves the way..." might appear too general. In order for the abstract to better focus on our findings we have removed it from the abstract.

Comment #2) There is no doubt that HHG should have some imprints on the distribution of IR photons. The question is how we decode this information. The authors’ method is not quite convincing to me. The main reason is that the anti-correlated signals [between PD1 and PMT signals -- see their Fig. 3(b)] might still be contaminated due to the excitation of the bound electrons to ATI electrons which correspond to the same number of photon absorption as the HHG process at any given harmonic. In fact, an electron that absorbs q photons might or might not recombine with the parent ion. My understanding is that the current experiment is still not a coincidence experiment so it can’t exclude those possibilities in an ensemble of a huge number of atoms. To partially clarify this issue I think the authors should provide a figure for PMT distribution to show what they actually measured.

Regarding the possible contamination of the correlation data, as has been described in the manuscript, the IR distribution is obtained after the correlation with XUV. This is done in order to minimize the contamination from ATI and all other processes. In order to make this issue more transparent, the sentence " S_{PD1} was used for recording the probability distribution of the energy of IR_1 , while S_{PD2} and S_{PMT} were used for removing the energy fluctuations of the laser

and the unwanted background, respectively." which appears at the end of the 1st paragraph of page 4 of the manuscript

has been replaced by

" S_{PD1} was used for recording the probability distribution of the energy of IR_1 , while S_{PD2} and S_{PMT} were used for removing the energy fluctuations of the laser and the unwanted background caused by processes irrelevant to the XUV emission, respectively. The background is an XUV-IR non-correlated signal which introduces an offset and reduces the visibility of the IR distribution which is correlated with the XUV emission."

Additionally, the graphs of the PMT distribution in case of Xenon and Argon have now been added in the Supplementary Information of the manuscript.

*Although the understanding of the XUV energy distribution requires further theoretical analysis for obtaining the wave function of the XUV radiation, we agree with the referee that the PMT distribution can further support our findings and also can be useful for the reader. For this reason, and for the sake of completeness, the graphs of the PMT distribution in case of Xenon and Argon have now been added in the Supplementary Information of the manuscript. Also, in order to address this issue in the main text of our manuscript the sentence **"Additionally to the above checks, the harmonic peak structure observed in the XUV distributions recorded by the PMT (Supplementary Information), and corresponding to Fig.3c and 3d, further supports the above findings."** has been added in page 8 of the main text of the manuscript.*

We hope that the harmonic peak structure shown in the XUV distribution, the additional information provided in the paper taking the advantage of the comments of the other referees, and the amount of systematic tests and measurements presented in our manuscript will relax the concerns of the referee.

Comment #3) There are few other technical details that are not clear to me. For example, the authors' estimate of IR photon number and the XUV photon number reduction is based entirely on XUV photon absorption. In reality, we know that HHG process depends sensitively on the macroscopic phase-matching condition. It is not quite clear if their method is able to detect this phase-matching effect in IR photon distribution. Another example is related to the contamination mentioned in (2). In fact, it is not quite clear to me if the method is capable of detecting, say, the Cooper minimum in HHG spectra from Ar.

The above comment has been divided in two parts (Comment #3a and #3b)

Comment #3a) For example, the authors' estimate of IR photon number and the XUV photon number reduction is based entirely on XUV photon absorption. In reality, we know that HHG process depends sensitively on the macroscopic phase-matching condition.

We agree that the HHG process depends on the macroscopic phase-matching conditions, but probably the referee missed that this point was written in the methods and Supplementary info of the manuscript. In the last paragraph of the "Supplementary information" was written that "Additionally, we note that although an accurate calculation of the probability distribution requires the consideration of the IR laser bandwidth and the propagation effects in the medium,

the fundamental properties of the interaction can be adequately explored with the single-color single-atom interaction, as has been done for the calculation of the XUV spectrum in the work of Lewenstein et al." Because of this, we wrote in the method section of the manuscript that the estimations of the IR photon number are rough. The sentence used was "On the estimation of the IR photon number: To roughly estimate the main features of the IR photon distribution.....".

In order to strongly address this issue in the main text of the manuscript the sentences "Although an accurate calculation of the probability distribution requires the consideration of the IR laser bandwidth and the propagation effects in the medium, a rough estimation which can provide an indicative value of the IR photons absorbed towards the harmonic emission and the main features of their distribution can be given..." and "Although the above estimations are rough, depict the feasibility of performing IR photon distribution measurements for revealing the high order harmonic spectrum using conventional detection techniques.", have been added at the beginning and the end of the 3rd paragraph of page 5 (please also see the reply to the comment #2c of the referee #2 and the beginning of the reply to the referee #3).

Comment #3b) It is not quite clear if their method is able to detect this phase-matching effect in IR photon distribution. Another example is related to the contamination mentioned in (2). In fact, it is not quite clear to me if the method is capable of detecting, say, the Cooper minimum in HHG spectra from Ar.

As the referee mentioned in his/her comment #1 this is a proof of principle experiment where the XUV spectrum was recorded by measuring the IR distribution. In a first such work exhibiting a new way of harmonic detection we clearly cannot address all possible variants and applications of the method.

We feel that discussing the effect of phase-matching conditions in the IR distribution and Cooper minima (which according to PRA 83, 023420 (2011) is taking place in photon energies (~ 50 eV) not detectable in the present work as the harmonic cut-off energy is at ≈ 32 eV) is out of the scope of this work and would divert the attention of the readers from the main point of the manuscript.

In conclusions, the current manuscript presents some interesting results and therefore it definitely deserves to be published in some form. However, based on the above arguments I do not think it qualifies for Nature Communications.

Reply to Referee #2

We thank the referee for his/her detailed report and the recognition that our approach is very intriguing and will possibly lead to new interesting developments. His/her comments were very useful for improving our manuscript. Below we provide the answers to all comments.

The manuscript "High order harmonics measured by counting the photons of the infrared driving laser pulse" by N. Tsatrafyllis and coworkers presents the experimental evidence of a new investigation method in attosecond science. The approach the authors present in their manuscript is very intriguing and will possibly lead to new interesting developments. For this reason I think that this work deserves

publication in Nature Communications. Nevertheless, I cannot recommend publication as it is for the following reasons:

Comment #1) I found the claims in the abstract and in the conclusions a pretentious. The authors write: “Here, we reveal the quantum-optical nature of the HOH generation process by recording the HOH spectrum without the need of XUV spectrometer.” I do not see how recording the HOH spectrum without the spectrometer automatically implies that one reveals the quantum-optical nature of the HOH. The authors should explain better what they mean with quantum-optical nature. After all, a harmonics spectrum is composed by peaks at an energy corresponding to precise multiples of the fundamental frequency. By energy conservation, this implies that every harmonic has taken a finite (odd) number of IR photons. What the authors show is the corresponding effect on the IR generating pulse. The two aspects seem to me to be linked since they represent two distinct ways to look at the same process. So it is not clear to me why one should not be able to conclude the quantum-optical nature of HHG already from the spectral distribution of the harmonics. On line 133, the sentence “These findings constitute an experimental demonstration of the quantum-optical nature of the strong-field light-atom interaction” pretends to be more general than what it should be. After all strong-field light-atom interaction describes a multitude of effects (e.g. above threshold ionization, tunneling ionization, HHG, ...) and the authors just investigated one of them. One line later the authors claim: “In particular, they allow the development of new high-photon-number non-classical light sources”. This theme is never addressed in the rest of the manuscript and I do not see how counting the IR photons which are anti-correlated to the XUV photons will help to develop a new non-classical light source. The authors should justify better their claim or remove it.

The above comment has been divided in three parts (Comment #1a,b,c)

Comment #1a) I found the claims in the abstract and in the conclusions a pretentious. The authors write: “Here, we reveal the quantum-optical nature of the HOH generation process by recording the HOH spectrum without the need of XUV spectrometer.” I do not see how recording the HOH spectrum without the spectrometer automatically implies that one reveals the quantum-optical nature of the HOH. The authors should explain better what they mean with quantum-optical nature. After all, a harmonics spectrum is composed by peaks at an energy corresponding to precise multiples of the fundamental frequency. By energy conservation, this implies that every harmonic has taken a finite (odd) number of IR photons. What the authors show is the corresponding effect on the IR generating pulse. The two aspects seem to me to be linked since they represent two distinct ways to look at the same process. So it is not clear to me why one should not be able to conclude the quantum-optical nature of HHG already from the spectral distribution of the harmonics.

This comment motivated us to further discuss this issue in the paper. It is true, that by energy conservation someone expects that the harmonic energy is taken by the IR field. The whole point is how one can measure this. Here, we do not spectrally resolve the radiations, but instead we measure only their energy (or photon number). Thus, is not straight forward how someone can measure the XUV spectrum by measuring the photon number of the IR radiation. To our knowledge, the only way to do this is to measure the probability distribution of the IR field exiting the harmonic generation medium. This is a purely quantum optical effect, which cannot be treated semi-classically. For this reason we have used the expression "Quantum-optical nature". In order to address this issue in the manuscript we have added the following sentence in

the last paragraph of page 2: "The quantum-optical nature of the measurement relies on the fact that we do not solely exhibit energy conservation in the IR-XUV interaction, which is an expected aggregate effect, but we explicitly measure photon number distribution, which has a quantum-optical nature."

Comment #1b) On line 133, the sentence "These findings constitute an experimental demonstration of the quantum-optical nature of the strong-field light-atom interaction" pretends to be more general than what it should be. After all strong-field light-atom interaction describes a multitude of effects (e.g. above threshold ionization, tunneling ionization, HHG, ...) and the authors just investigated one of them.

We agree with the referee. This sentence might indeed be too general, and for this reason we have modified in the following way "These findings constitute an experimental demonstration of the quantum-optical nature of the high order harmonic generation process".

Comment #1c) One line later the authors claim: "In particular, they allow the development of new high-photon-number non-classical light sources". This theme is never addressed in the rest of the manuscript and I do not see how counting the IR photons which are anti-correlated to the XUV photons will help to develop a new non-classical light source. The authors should justify better their claim or remove it.

Again we agree with the referee. This statement of ours might indeed be premature, hence this sentence has been removed.

Comment #2) I found the description of the origin of the IR photon number at page 3 and 4 badly written and confusing. There are quantities that are not properly introduced. Let me be more precise. At line 59 the authors write: "it is found that the IR photon number probability distribution after the medium depends on the probability amplitudes A_n and a phase Φ_n ". These two quantities (A_n and Φ_n) neither are introduced explicitly in the manuscript nor in the supplementary. So, without reading Ref. [10], this sentence conveys no message if not that the probability depends on its amplitude and its phase. This is obvious and true for any quantity. The quantity N_0' at line 77 has not been defined. One can guess that it is the photon number when the HHG has been turned on, but this should be clearly stated. Lines 82 to 85 are difficult to follow. At this point of the manuscript the reader is left with the impression that a voltage signal on the photodiode magically transforms in photon counts. How this happens is totally unclear. Are the photodiodes calibrated or not? How is the XUV photon number ($\sim 10^8$ photon/pulse) measured? How one can estimate from that number that we have $N_q(\text{IR})$ of 2×10^{13} ? And what is the quantity T ? It is true that the authors refer to the method section where these quantities are explained, but this part is crucial to understand the whole work and thus should be described better already in the manuscript. In particular because if I take the quantities as they are written in the manuscript I get to an unphysical result. If I take $\Delta N_q(\text{IR}) \sim 10^7$ (line 84), $N_q(\text{IR}) \sim 2 \times 10^{13}$ (line 83) and the definition of R at line 88, I find that, in order to have $R=2$, q should be 4×10^7 which does not make sense. Can the problem be that $\Delta N_q(\text{IR})$ at line 84 is not the real one, but just the one transmitted through the filter? I have also found another apparent inconsistency in the method section. At line 193 $N_q(\text{IR}) \sim 2 \times 10^{13}$ is associated to an expected difference of $\Delta N_q(\text{IR}) = 4 \times 10^{12}$. But if I take the definition of $N_q(\text{IR}) = A_q N_q(\text{XUV})$ I should expect that $\Delta N_q(\text{IR}) = 2 A_q N_q(\text{XUV})$. If I put in the values for A and $N_q(\text{XUV})$ I get $\Delta N_q(\text{IR}) = 5 \times 10^4 \square 2 \times 10^7 \square 2 = 2 \times 10^{12}$ which is half of what reported at line 195.

Finally, I find confusing the definition of $N_q(\text{XUV})$. In figure 3 the authors clearly show that the harmonic intensity is not uniform. So I do not expect $N_q(\text{XUV})$ to be constant, but to vary significantly with q . To what does the value 2×10^7 correspond? Is it an average or the value for the harmonics in the plateau?

The above comment has been divided in six parts (Comment #2a,b,c,d,f)

Comment #2a) At line 59 the authors write: "it is found that the IR photon number probability distribution after the medium depends on the probability amplitudes A_n and a phase Φ ...". These two quantities (A_n and Φ) neither are introduced explicitly in the manuscript nor in the supplementary. So, without reading Ref. [10], this sentence conveys no message if not that the probability depends on its amplitude and its phase. This is obvious and true for any quantity.

Probably the referee missed the introduction of these quantities. They were introduced in the second paragraph of page 2 of the supplementary info. In order to address this we have added "(Supplementary Information)" at the end of the sentence "In particular [10], it is found that the IR photon number probability distribution P_n after the medium depends on the probability amplitudes A_n and a phase $\Phi(t_i, t_r)$ (t_i , t_r are the ionization and recombination times, respectively)" which appears at the end of page 3 of the manuscript.

Comment #2b) The quantity N_0' at line 77 has not been defined. One can guess that it is the photon number when the HHG has been turned on, but this should be clearly stated.

That's correct, we thought it was self-evident, but in the new version of the manuscript we have added the sentence " N_0' reflects the remained IR photon number resulted after the absorption of $N_q^{(IR)}$ and $N_{abs}^{(IR)}$ photons."

Comment #2c) Lines 82 to 85 are difficult to follow. At this point of the manuscript the reader is left with the impression that a voltage signal on the photodiode magically transforms in photon counts. How this happens is totally unclear. Are the photodiodes calibrated or not? How is the XUV photon number ($\sim 10^8$ photon/pulse) measured? How one can estimate from that number that we have $N_q(\text{IR})$ of 2×10^{13} ? And what is the quantity T ? It is true that the authors refer to the method section where these quantities are explained, but this part is crucial to understand the whole work and thus should be described better already in the manuscript.

Indeed, it is a good idea to include this discussion in the main manuscript. In order to make a smooth transition from the "voltage signal" to "photon number" we have added the following text in page 5: "The calibration of the PMT signal to the photon number was done by taking into account the quantum efficiency of the detector, the reflectivity of the XUV optics and the XUV filter transmission (Method Section). In case of IR, the signal of the diodes was calibrated to the photon number using three different approaches which lead to the same result: I) by corresponding the diode signal to the IR energy value (measured by a power-meter) taking into account the transmission of the optical elements and the neutral density filters (with transmission coefficient T , with $T^{-1} = 3 \times 10^6$) which have been used in order to avoid saturation effects in the diode, II) using the specifications (responsivity and load resistance) of the photodiodes and III) by means of a single photon counter."

Also, now a large part from the method section moved in the main text in a way that the reader can consistently reproduce the numbers, i.e. in the revised version of the manuscript the following text "Taking into account the measured XUV photon harmonic emission (methods section)."

has been replaced by

" Although an accurate calculation of the probability distribution requires the consideration of the IR laser bandwidth and the propagation effects in the medium, a rough estimation which can provide an indicative value of the IR photons absorbed towards the harmonic emission and the main features of their distribution can be given by correcting the measured harmonic photon number for the XUV absorption effects. The XUV photon number at the output of the Xenon gas found to be in the order of $N^{(XUV)} \sim 10^8$ photons/pulse translating to $N_q^{(XUV)} \approx (N^{(XUV)}/5) \approx 2 \times 10^7$ photons/harmonic (where 5 is the number of harmonics lying in the plateau region). Taking into account the experimental conditions, it turns out that the XUV photon number is reduced due to absorption effects in the medium by a factor of $A \approx 5 \times 10^4$ (Method section). Considering that q IR photons are required for the generation of the q th harmonic, the number of the IR photons absorbed towards harmonic emission is $\tilde{N}_q^{(IR)} = AqN_q^{(XUV)} \approx 2 \times 10^{13}$ photons/pulse (for $q = 17$), translating to $N_q^{(IR)} = \tilde{N}_q^{(IR)}T \sim 10^7$ photons/pulse at PDI. Additionally, the photon number difference between consecutive peaks ($\Delta q=2$) in the IR_1 probability distribution is expected be $\Delta N_q^{(IR)} = \tilde{\Delta N}_q^{(IR)}T \sim 10^6$ photons/pulse (where $\tilde{\Delta N}_q^{(IR)} \approx A(\Delta q)N_q^{(XUV)} \approx 2 \times 10^{12}$ photons/pulse). Although the above estimations are rough, depict the feasibility of performing IR photon distribution measurements for revealing the high order harmonic spectrum using conventional detection techniques. Additionally, we would like to point out that, although the interpretation of our measurement is independent of the value of n_a , the derivation of n_a from the detected number of XUV/IR photons requires careful consideration of the experimental conditions and propagation effects of the XUV/IR fields in the harmonic generation medium [11, 12] which is beyond the scope of this work.

Taking into account the above, S_{PDI} is expected to have a distribution located around $N'_0 \approx 2 \times 10^8$ photons/pulse with the "harmonic-peak" structure being about 2 orders of magnitude smaller. A quantitative parameter which can be used for the justification of the validity of the measurements is the parameter $R = \frac{q\Delta N_q^{(IR)}}{N_q^{(IR)}} = \Delta q$ (for consecutive harmonics should be $R=2$). Experimentally this parameter can be obtained by subtracting the number of the IR photons absorbed due to processes other than harmonic emission ($N_{q=0}^{(IR)}$) (Supplementary Information)."

Also, a part of experimental procedure which includes the definition of the transmission coefficient T , has now been moved from the method section in the main text of the manuscript.

Comment #2d) In particular because if I take the quantities as they are written in the manuscript I get to an unphysical result. If I take $\Delta N_q^{(IR)} \sim 10^7$ (line 84),

$N_q(\text{IR}) \sim 2 \times 10^{13}$ (line 83) and the definition of R at line 88, I find that, in order to have $R=2$, q should be 4×10^7 which does not make sense. Can the problem be that $\Delta N_q(\text{IR})$ at line 84 is not the real one, but just the one transmitted through the filter?

Indeed, perhaps the way the text was written was confusing and leading to miscalculations. To clarify, for the calculation of R we use the measured values of $\Delta N_q^{(\text{IR})}$ and $N_q^{(\text{IR})}$. This is now described in a transparent way in the section "Obtaining the $N_{q=0}^{(\text{IR})}$ and parameter R " of the supplementary information of the paper by the sentence "The parameter R was obtained through the relation

$$R = \frac{q(N_{q=0}^{(\text{IR})} - N_{q-1}^{(\text{IR})}(\text{meas.})) - (N_{q=0}^{(\text{IR})} - N_{q+1}^{(\text{IR})}(\text{meas.}))}{N_{q=0}^{(\text{IR})} - N_q^{(\text{IR})}(\text{meas.})} = \frac{q(N_{q+1}^{(\text{IR})}(\text{meas.}) - N_{q-1}^{(\text{IR})}(\text{meas.}))}{N_{q=0}^{(\text{IR})} - N_q^{(\text{IR})}(\text{meas.})},$$

where $N_q^{(\text{IR})}(\text{meas.})$ is the measured photon number corresponding to the q th "IR-harmonic" peak and $N_{q-1}^{(\text{IR})}(\text{meas.})$, $N_{q+1}^{(\text{IR})}(\text{meas.})$ are the measured photon numbers corresponding at the left and right minima of the $N_q^{(\text{IR})}(\text{meas.})$ value, respectively."

If someone wants to roughly calculate R taking into account the numbers obtained by the rough estimations i.e. $\Delta N_q^{(\text{IR})} \sim 10^6$, $N_q^{(\text{IR})} \sim 10^7$ (which they do not consider the dependence of $\Delta N_q^{(\text{IR})}$ and $N_q^{(\text{IR})}$ on q) R takes values in a physically acceptable range (depending on the harmonic order ($q=9-23$)). We have also included this comment in the Supplementary Information of the manuscript.

Comment #2e) I have also found another apparent inconsistency in the method section. At line 193 $N_q(\text{IR}) \sim 2 \times 10^{13}$ is associated to an expected difference of $\Delta N_q(\text{IR}) = 4 \times 10^{12}$. But if I take the definition of $N_q(\text{IR}) = AqN_q(\text{XUV})$ I should expect that $\Delta N_q(\text{IR}) = 2AN_q(\text{XUV})$. If I put in the values for A and $N_q(\text{XUV})$ I get $\Delta N_q(\text{IR}) = 5 \times 10^4 \times 2 \times 10^7 \times 2 = 2 \times 10^{12}$ which is half of what reported at line 195.

We thank the referee once again for the careful reading. We agree. This was a typo, which now has been corrected. The correct value is $\widetilde{\Delta N}_q^{(\text{IR})} = A(\Delta q)N_q^{(\text{XUV})} \approx 2 \times 10^{12}$ photons/pulse.

Comment #2f) Finally, I find confusing the definition of $N_q(\text{XUV})$. In figure 3 the authors clearly show that the harmonic intensity is not uniform. So I do not expect $N_q(\text{XUV})$ to be constant, but to vary significantly with q . To what does the value 2×10^7 correspond? Is it an average or the value for the harmonics in the plateau?

This value corresponds to the plateau harmonics and has been obtained by dividing the total XUV photon number by the number of the harmonics lying in the plateau i.e. $N_q^{(\text{XUV})} \approx N^{(\text{XUV})}/5$. In order to clarify this in the manuscript, the text " i.e. $N_q^{(\text{XUV})} \sim 2 \times 10^7$ photons/pulse" in the method section has been replaced by "i.e. $N_q^{(\text{XUV})} \approx (N^{(\text{XUV})}/5) \sim 2 \times 10^7$ photons/harmonic (where 5 is the number of harmonics lying in the plateau region)". This is also added in the main text of the manuscript.

Comment #3) At line 120 the authors write: "'IR-harmonics" intensity is consistent with the spectrum measured with the XUV spectrometer (blue dots in Fig. 3d).", but in

figure 3 I do not see the full XUV spectrum. For the sake of a better comparison and comprehension, the authors should show the full UXV spectrum instead of the blue dots in fig. 3c and 3d.

The blue dots in Fig. 3c and 3d have been replaced by the full XUV spectrum.

Comment #4) I couldn't find any information on the laser repetition rate.

We agree. The rep. rate of the laser was 10Hz. This is now written in the main text of the manuscript.

Comment #5) Is the part on the calibration of the IR-harmonic spectrum in the methods implying that one anyway needs the XUV spectrum to calibrate the signal? If so, this should be openly stated in the manuscript too.

In principle the measurements of the XUV spectrum is not needed. Here, in a proof of principle experiment, we recorded the XUV in order to confirm the validity of the IR distribution measurements.

Comment #6) Minor comments:

- line 92, Fig 3A is marked with capital letter while all the others are labeled with small letter.
- In fig. 4 the dots are difficult to read. I suggest to make the lines dashed instead of dotted.

Corrected

Reply to Referee #3

We thank the referee for the detailed report. His/her comments were very useful for improving our manuscript. Below we provide the answers to all comments of the referee.

Before addressing one by one the comments of the referee we would like to make a general comment concerning the estimations of the IR photon number. These estimations are meant to give a rough estimation of the IR photons absorbed in order only to show that the quantity of the IR photons leading to XUV emission can be measured with conventional techniques.

In order to avoid further confusion on this matter, this point has been now strongly addressed in the main text of the manuscript by using the following two sentences in the last paragraph of page 5-beginning of page 6,

Last paragraph of page 5: "Although an accurate calculation of the probability distribution requires the consideration of the IR laser bandwidth and the propagation effects in the medium, a rough estimation which can provide an indicative value of the IR photons absorbed towards the harmonic emission and the main features of their distribution can be given by correcting the measured harmonic photon number for the XUV absorption effects."

Beginning of page 6: "Although the above estimations are rough, depict the feasibility of performing IR photon distribution measurements for revealing the high order harmonic spectrum using conventional detection techniques."

The authors measure the probability distribution of the energy of an ultrashort laser pulse after it has undergone intense nonlinear propagation in a gas, producing high-order harmonics. The probability distribution, conditioned on simultaneous measurements of the energy of the harmonics, appears similar to the spectrum of the harmonics measured on a conventional XUV spectrometer. The authors' interpretation is that this is due to the quantum optical nature of the interaction, which quantizes the number of photons absorbed per gas atom. I congratulate the authors on their attempt at tackling an interesting unsolved problem - the quantum optical nature of HHG - in a creative and original way. However I have some significant concerns with the concepts and interpretation, as well as the details of the data processing which are crucial links in the authors chain of reasoning. For this reason I do not support publication of the manuscript in its present form, but would re-evaluate my opinion if significant improvements were made which addressed these problems. The manuscript relies heavily on the concept of the number of atoms with which the light interacts coherently, n_a . This needs more explanation - as presented, it has some serious contradictions with existing knowledge.

Comment #1) The authors state that each atom absorbs q IR photons towards the generation of the q -th order harmonic, and as such the number of missing IR photons is $N_q^{(IR)} = qn_a$. This contradicts well established theory (see e.g. Eberly et al. PRA 1992 45 p4706) that the intensity of light forward scattered coherently by a collection of atoms scales as the number of atoms squared. This fact is widely known and built into all macroscopic models of HHG, and is experimentally well established. In the authors' picture, if I have a system with $N_q^{(IR)}$ photons contributing to the q th harmonic, and then I double the number of atoms, then I now have $2N_q^{(IR)}$ photons contributing. The standard viewpoint says I will have 4 times the intensity and hence four times the number of XUV photons. Where do the extra photons come from?

Comment #2) I suspect, but am not certain, that the resolution of these conflicting views lies in the fact that the experiment is conducted in the absorption saturated regime. The authors state that the absorption length is much smaller than the target thickness. They do not discuss the coherence length, but it is implied to be much longer. In the absorption saturated regime (see e.g. Constant et al. PRL 82 p1668 1999), the number of harmonic photons per unit cross sectional area is independent of the atomic density, but if the area is scaled then the total number of harmonic photons could scale as the number of atoms. (This is just one possibility...)

We thank the referee for the insightful comment. The way the text was written was confusing. We would like to point out that the description and the explanation of the present results is entirely independent on the measurement of n_ω , and thus n_a has been removed from any discussion and analysis of the experimental data. Please, see also the comment #2c of the referee #2 where the number n_a has been removed from the estimations presented in page 5-6 of the main text of the manuscript.

In order to further clarify this point and the same time to make the connection between the experiment and the theory we have now added at the beginning of page 6 of the main text of the manuscript the sentence "Additionally, we would like to point out that, although the

interpretation of our measurement is independent of the value of n_a , the derivation of n_a from the detected number of XUV/IR photons requires careful consideration of the experimental conditions and propagation effects of the XUV/IR fields in the harmonic generation medium [Eberly et al. PRA 45, 4706 (1992); Constant et al. PRL 82, 1668 (1999)] which is beyond the scope of this work." (please see the revised text in the reply to the comment #2c of the referee #2).

Additionally the sentence " Note, that n_a is a parameter which can be obtained by the detected photon number taking into account the propagation effects and phase matching conditions [Eberly et al. PRA 45, 4706 (1992); Constant et al. PRL 82, 1668 (1999)]. " has been added in page 3 of the main text of the manuscript and in the supplementary information of the manuscript.

Also we agree with the referee regarding the comment about the "XUV absorption saturated regime". What the referee is suspected is indeed correct. The experiment is conducted in the absorption saturated regime. In order to clarify this issue in the methods section of the manuscript the text

" For our experimental conditions where $L_{med} \approx 1.2$ mm, the XUV absorption length caused by single-XUV-photon-ionization process $L_{abs}^{(XUV)} = 1/\rho\sigma^{(1)} \approx 100\mu\text{m}$ (cross section of Xenon $\sigma^{(1)} \approx 3 \times 10^{-17} \text{ cm}^2$) it follows that XUV photons are reduced due to absorption along over all the medium length (integration over the medium length) by a factor of $A \approx 5 \times 10^4$."

has been replaced by

" For our experimental conditions where $L_{med} \approx 1.2$ mm, the XUV absorption length caused by single-XUV-photon-ionization process is $L_{abs}^{(XUV)} = 1/\rho\sigma^{(1)} \approx 100\mu\text{m}$ (cross section of Xenon $\sigma^{(1)} \approx 3 \times 10^{-17} \text{ cm}^2$). As $L_{med} \gg L_{abs}^{(XUV)}$ and $L_{coh} \gg L_{abs}^{(XUV)}$ (where $L_{coh} = \pi/\Delta k$ is the coherent length [A. Rundquist et al., Science 280, 1412 (1998); E. A. Gibson et al., Science 302, 95 (2003)]], $\Delta k = k_L - qk_L$ and k_L is the wave number of the fundamental) it follows that the present experiment is conducted in the XUV absorption saturation regime. In this context the XUV photon number reaches the value of $N^{(XUV)} \sim 10^8$ photons/pulse at the beginning of the medium and remains constant along the propagation as the XUV losses induced by the single photon ionization are getting balanced by the IR absorption which takes place along the propagation in the whole medium. Considering the medium as a single absorbing filter having exponential dependence on the medium length it follows that XUV photons are reduced due to absorption along over all the medium length (integration over the medium length) by a factor of $A \approx 5 \times 10^4$. Considering that q IR photons are required for the generation of the q th harmonic, the number of the IR photons absorbed towards harmonic emission is $\tilde{N}_q^{(IR)} = AqN_q^{(XUV)} \approx 2 \times 10^{13}$ photons/pulse (for $q = 17$)."

Comment #3) If we accept the authors statement that each atom absorbs q photons to make one XUV harmonic, then n_a must be harmonic order dependent, or else the harmonic spectrum would be flat. What then does n_a really mean – the number of atoms in a volume of sufficient laser intensity to produce a given harmonic? This idea should be more carefully introduced.

We feel that there is a misunderstanding here. The relative amplitude distribution of the "IR-harmonic" peaks depends on the recollision process at the single-atom level, and not on the number of atoms participating in the XUV emission. This is shown in the theoretical calculations in upper-right panel of Fig.1 which have been performed for a q -independent $n_a=500$ atoms. The number of atoms effects only the width and the spacing of the peaks. A q -dependant n_a can influence the width of the "IR-harmonic" peaks and lead to deviations of R form the value of 2. Such effects are not observable in our case due to the finite resolution of the measurement, and thus we did not explicitly take them into account.

In order to address this issue, the text "We note that spatial intensity distribution and propagation effects in the harmonic generation medium can cause subtle dependence of n_a on q . Although such effects can influence the width of the "IR-harmonic" peaks and lead to deviations of R form the value of 2, they have not been taken into account in the present work as they were not observable due to the finite resolution of the measurements." have been added at the end of page 4 of the Supplementary Information of the manuscript.

In additional to these conceptual concerns, I found some of the data processing hard to follow, with important details missing.

Comment #3.1) I couldn't precisely follow all steps in the derivation of R . The authors state that the spacing of $\Delta N_q(\text{IR})$ is proportional to $2Tn_a$. If we assume equality rather than proportionality was intended, then the definition of R follows from substituting $N_q(\text{IR})=qn_a$. This should be made clear.

*We agree with the referee. This issue has been corrected.
Please see the reply to the comment #2d of the referee #2.*

Comment #3.2) The procedure for obtaining $N_q(\text{IR})$ is described in the last subsection of the Methods section. In the description, it is not clear which values are resolved by harmonic, and which are totals. It seems that $N(\text{XUV})$ is the total detected harmonic photon number, but the gas absorption used to calculate the emission from the gas must have some dependence on frequency. Then, the quantity $N_q(\text{XUV})$ appears without introduction. Is this the XUV photon number resolved by harmonic? Do the authors use the measured spectrum to split the total number of XUV photons into individual harmonics?

*We agree again with the referee. This issue has been corrected.
Please see the reply to the comment #2c and #2f of the referee #2.*

Comment #3.3) It is stated that $N_{\{q=0\}}(\text{IR})$ is inferred from extrapolating the linear dependence of $N_q(\text{IR})$ on q . I didn't understand this. Partly the problem is notation - is $N_q(\text{IR})$ and experimental observable, in which case the formula $N_q(\text{IR})=AqN_q(\text{XUV})$ is incorrect and needs an additional term? Or is this formula correct, in which case $N_{(q=0)}(\text{IR})$ is identically zero.

We would like to thank the referee once more for the careful reading. The formula is correct, and $N_{(q=0)}(\text{IR})$ is indeed identically zero, since there is no harmonic produced at $q=0$, and thus there are no IR missing photons correlated with XUV. This now can be seen in the revised Fig. 3c and 3d, by comparing the IR harmonic distribution with the XUV spectrum recorded by the spectrometer. The value of $N_{(q=0)}(\text{IR})$ coincides with the value of $q=0$ obtained by the

harmonic spectrum (blue x-axes in Fig. 3c and 3d). This issue is explicitly addressed in the revised version by adding the sentence "We note the $N_{q=0}^{(IR)}$ coincides with the value of $q=0$ obtained by the harmonic spectrum (blue x-axes in Fig. 3c and 3d). This constitutes a further justification of the validity of the measurements." at the beginning of page 8 of the revised version of the paper and the Methods section.

Comment #3.4) The method of estimating the number of XUV photons isn't described in sufficient detail. The authors speak of the XUV photons being reduced by absorption along the medium length. How is this modelled? Do they treat the harmonics as being generated without absorption, and then after generation passing through the gas, modelled as a single filter? Or do they model the simultaneous buildup and absorption of harmonics, which (provided there is phase matching) reaches an equilibrium after several absorption lengths (see e.g. Constant et al. PRL 1999 82). In this case, the number of photons predicted without absorption would be very large, as coherent build-up of the field would produce a signal proportional to the number of atoms squared. This process is crucial, since it is used in evaluating R.

This issue has been now addressed in our reply to the comments #1 and #2.

Comment #3.5) The superimposed XUV spectra seem to agree nicely. With the caveats described above, I was able to understand how the authors scaled the measure photon number to give harmonic order. What about the offset? Apart from maximizing the similarity of the XUV spectra and photon number distribution, is there any justification for assigning the peak of the photon number distribution to a certain harmonic (e.g. harmonic 15 in Fig. 3c).

Please see the reply to the comment #4c. Further information regarding the calibration of the "IR-harmonic" spectrum is described in detail in the method section with title " On the calibration of the "IR-harmonic" spectrum".

Comment #3.6) Are all the arbitrary units the same throughout for S_PD1, S_PD2, S_PMT i.e. can numbers on the vertical axis of Fig. 20 be compared directly to those on the vertical axis of Fig. 3(b) (and likewise for all other plots of the S_ signals)?

In principle this is something that can be done, but because the measurement of Fig 2c is meant only to show the dependence of the signals on intensity of the IR and the laser intensity area where the IR distribution measurements have been performed, we would like to avoid the direct comparison between the figures 2c and 3b.

Comment #3.7) How were the photodiodes, particularly PD1, calibrated? This is crucial information.

This information is now added in the main text of the manuscript by using the sentence: At the 2nd paragraph of page 5 of the main text we have added the sentence: " The calibration of the PMT signal to the photon number was done by taking into account the quantum efficiency of the detector, the reflectivity of the XUV optics and the XUV filter transmission (Method Section). In case of IR, the signal of the diodes was calibrated to the photon IR number using three different approaches which lead to the same result: I) by corresponding the diode signal to the IR energy value (measured by a power-meter) taking into account the transmission of the optical elements and the neutral density filters (with transmission

coefficient T with $T^{-1}=3 \times 10^6$) which have been used in order to avoid saturation effects in the diode, II) using the specifications (responsivity and load resistance) of the photodiodes and III) by means of a single photon counter."

Reviewers' comments:

Reviewer #1 (Remarks to the Author):

The authors have addressed all my comments/suggestions. I found their responses quite satisfactory and the revised version has been much improved. I now recommend the paper for publication in Nature Communications.

Reviewer #2 (Remarks to the Author):

Article ID: NCOMMS-16-21519-T

Title: High order harmonics measured by counting the photons of the infrared driving laser pulse

The manuscript "High order harmonics measured by counting the photons of the infrared driving laser pulse" by N. Tsatrafyllis and coworkers has been carefully revised and it is now easier to read. I recommend this work for publication in Nature Communications after the authors clarify these last two points:

1) At line 126 and in the following paragraph the authors talk about the parameter R. They introduce this parameter by saying: "A quantitative parameter which can be used for the justification of the validity of the measurements is the parameter R, ... (for consecutive harmonics should be $R=2$).". From this the reader understands that R is a quite important parameter that can be used to verify the validity of the measurements. So, as a reader, I expected to see in the paper (or in the supplementary) what was the value that the authors obtained for R. Otherwise I do not see why R should be even introduced. Unfortunately, I did not find the values of R in the paper or in the supplementary. In the supplementary, lines 103-105, the authors invite the reader to calculate R on their own and state that the reader will find meaningful results. If it is true that R is so important to deserve to be mentioned in the paper, I think the authors should add their estimate of this parameter and comment why it justifies the validity of their measurements.

2) As Referee#1 correctly pointed out, the IR spectrum could be still contaminated by other processes. In particular, ATI will give the same IR photon absorbed, but no HOH generation. In the present form of the paper, the authors try to address this concern from Referee#1, saying that there are processes irrelevant to the XUV emission. I think that it will be useful to add the value of the Keldish parameter γ for their experiment to the discussion. It should show that the ionization happens in the tunneling regime and not in the multi-photon, thus giving an idea on the strength of the ATI process.

Reviewer #3 (Remarks to the Author):

I appreciate the authors' changes to the manuscript, and still maintain my original positive opinion of the novelty and interest of the work. However, in the revised manuscript I still find much that is contradictory or confusing. Furthermore, I suspect that the fact that $R=2$ (which is a key piece of evidence supporting their conclusion - see line 168) is inevitable given their data processing procedure, and therefore not compelling evidence of the central claims. Therefore I cannot support publication of the manuscript in its present form. Some detailed comments follow.

In response to my comment 3.3, the authors state that $N_{(q=0)}^{(IR)}$ is identically zero. Yet in the supplementary information around Fig. SI.1, they describe their extrapolation procedure which leads

to a value of $N_{(q=0)}^{(IR)} = 2.1 \times 10^8$ photons/pulse! This also appears in Fig. 3(c). On line 131, it is stated that $N_{(q=0)}^{(IR)}$ is the number of IR photons absorbed due to processes other than harmonic emission. All of this would seem to contradict the authors' response. Furthermore, on line 91 it is stated that $N_{abs}^{(IR)}$ is the number of photons of the IR absorbed towards processes other than harmonic emission. Are $N_{abs}^{(IR)}$ and $N_{(q=0)}^{(IR)}$ the same?

I wonder if some of this confusion is caused by mixing up <<detected>> photon numbers, which seem to be denoted on the plots by lower case n, with the <<derived>> photon numbers that appear in the text as a capital N. An example of a derived photon number (at least, according to the given definition), is $N_q^{(IR)}$, defined on line 91 as the number of IR photons absorbed towards harmonic generation. However, to make sense of Fig SI.1. one has to treat $N_q^{(IR)}$ as a detected photon number, read straight off the vertical axis. To put it very plainly: does $N_q^{(IR)}$ increase with q, as the equations on lines 115 and 116 imply, or does it decrease with q, as Fig SI.1. shows?

Due to the aforementioned issues, I am still not completely certain of the procedure used to process the data and obtain $N_q^{(IR)}$ (for nonzero q and q=0), and hence R. However, it seems to me that in performing the linear extrapolation to obtain $N_{(q=0)}^{(IR)}$ (Fig. SI.1.), the authors have implicitly assumed that $R=2$, so that this fact does not serve as a 'check' to validate the spectrum. To put it in more detail: as I understand it, authors measure a distribution of detected IR photon numbers $f(n)$ which contains regularly spaced peaks. Based on comparison to the conventional harmonic spectrum, they assign to the peaks consecutive odd integers q representing harmonic orders. They fit a straight line vs q to the position of the peaks and extrapolate to q=0 to obtain $n_{(q=0)}$. They define $N_q^{(IR)} = n_{(q=0)} - n_q$, the difference in position between a peak and the extrapolated q=0 value. If the linear extrapolation is good, then it follows (e.g. from similar triangles) that $\Delta N_q^{(IR)} / 2 = N_q^{(IR)} / q$ and hence R (as defined on line 129) equals 2 - regardless of the actual experimental data!

Line 119: Does $A(\Delta q)$ mean $A \cdot \Delta q$ or A evaluated at Δq ? If the latter, what does this mean?

Reply to referees for the manuscript (Nr.: NCOMMS-16-21519A) with title "*High order harmonics measured by counting the photons of the infrared driving laser pulse*" submitted to Nature Comm. by N. Tsatrafyllis et al.

Our reply to the referees comments are written in blue and italic.

Reply to Referee#1

The authors have addressed all my comments/suggestions. I found their responses quite satisfactory and the revised version has been much improved. I now recommend the paper for publication in Nature Communications.

We would like to thank the referee for recommending our manuscript for publication in Nature Comm.

Reply to Referee #2

The manuscript "High order harmonics measured by counting the photons of the infrared driving laser pulse" by N. Tsatrafyllis and coworkers has been carefully revised and it is now easier to read. I recommend this work for publication in Nature Communications after the authors clarify these last two points:

We would like to thank the referee for recommending our manuscript for publication in Nature Comm. Please see below the corresponding modifications, which have been done in the manuscript in order to clarify these two points.

Point #1) At line 126 and in the following paragraph the authors talk about the parameter R . They introduce this parameter by saying: "A quantitative parameter which can be used for the justification of the validity of the measurements is the parameter R , ... (for consecutive harmonics should be $R=2$).". From this the reader understands that R is a quite important parameter that can be used to verify the validity of the measurements. So, as a reader, I expected to see in the paper (or in the supplementary) what was the value that the authors obtained for R . Otherwise I do not see why R should be even introduced. Unfortunately, I did not find the values of R in the paper or in the supplementary. In the supplementary, lines 103-105, the authors invite the reader to calculate R on their own and state that the reader will find meaningful results. If it is true that R is so important to deserve to be mentioned in the paper, I think the authors should add their estimate of this parameter and comment why it justifies the validity of their measurements.

The discussion about R was meant to show that using the calibration procedure described in the manuscript the measured spacing between the consecutive "IR-harmonic" peaks is constant and in agreement with the expected value of consecutive harmonics. In other words, R describes the quality of the linear dependence of Nq versus q , i.e. the fact that the spacing between consecutive IR harmonic peaks is constant.

Nevertheless, we infer from the comment of the referee that the discussion of R is more confusing rather than illuminating, so we opt to remove it and just mention explicitly the constant spacing between consecutive peaks.

Point #2) As Referee#1 correctly pointed out, the IR spectrum could be still contaminated by other processes. In particular, ATI will give the same IR photon absorbed, but no HOH generation. In the present form of the paper, the authors try to address this concern from Referee#1, saying that there are processes irrelevant to the XUV emission. I think that it will be useful to add the value of the Keldysh parameter γ for their experiment to the discussion. It should show that the ionization happens in the tunneling regime and not in the multi-photon, thus giving an idea on the strength of the ATI process.

The values of γ have been added in the caption of Figure 3 using the sentence " We note that, as in the majority of the HHG experiments [12], in both gases (Xe, Ar) the intensity of the laser in the HHG regime was kept just below the ionization saturation intensity i.e. $\lesssim 10^{14}$ W/cm² where the value of the Keldysh parameter is $\gamma = (IP/2U_p)^{1/2} \approx 1$ (U_p is the ponderomotive energy of the electron). This value is typical for the majority of the HOH generation experiments where the ionization of an atom falls in the tunneling regime."

Reply to Referee #3

I appreciate the authors' changes to the manuscript, and still maintain my original positive opinion of the novelty and interest of the work. However, in the revised manuscript I still find much that is contradictory or confusing. Furthermore, I suspect that the fact that R=2 (which is a key piece of evidence supporting their conclusion - see line 168) is inevitable given their data processing procedure, and therefore not compelling evidence of the central claims. Therefore I cannot support publication of the manuscript in its present form. Some detailed comments follow.

We thank the referee for the recognition of the novelty of our work and his/her comments which helped us to improve the clarity and readability of the manuscript, which we feel we have now addressed to the fullest possible extent. Please see below the corresponding modifications, which have been done in the manuscript in order to clarify the confusing issues.

Comment #1) In response to my comment 3.3, the authors state that $N_{(q=0)}^{(IR)}$ is identically zero. Yet in the supplementary information around Fig. SI.1, they describe their extrapolation procedure which leads to a value of $N_{(q=0)}^{(IR)}=2.1 \times 10^8$ photons/pulse! This also appears in Fig. 3(c). On line 131, it is stated that $N_{(q=0)}^{(IR)}$ is the number of IR photons absorbed due to processes other than harmonic emission. All of this would seem to contradict the authors' response. Furthermore, on line 91 it is stated that $N_{abs}^{(IR)}$ is the number of photons of the IR absorbed towards processes other than harmonic emission. Are $N_{abs}^{(IR)}$ and $N_{(q=0)}^{(IR)}$ the same?

We agree, this point may lead to confusion. Please find below the change that we have done in page 7-8 of the manuscript, which clarifies this issue.

The sentence:

"Due to the weaker ionization of Argon $N_{q=0}^{(IR)} \approx 3 \times 10^8$ photons/pulse is closer to N_0 compared to the Xenon case ($N_{q=0}^{(IR)} \approx 2.1 \times 10^8$ photons/pulse), while as expected for HOH generation in Argon the number of "IR-harmonic" peaks was increased and the cut-off position moved to higher harmonics."

has been replaced by:

"The green-dashed line (noted as $N_{q=0}^{(IR)}$) is positioned at the number of measured IR photons n corresponding to $N_{q=0}^{(IR)}$, which is obtained by extrapolating to $q=0$ the linear dependence of $N_q^{(IR)}$ on q (Supplementary Information). This value of n coincides with the value corresponding to $q=0$ obtained from the harmonic spectrum (blue x-axis in Fig. 3c and 3d) and reflects the remained IR photon number resulting from the absorption due to processes other than harmonic emission ($N_{abs}^{(IR)}$). At the green dashed line, $N_{q=0}^{(IR)} = 0$ and the measured photon number (upper red x-axis) is $N_0 - N_{abs}^{(IR)}$. Due to the weaker ionization of Argon the position of $N_{q=0}^{(IR)}$ (where $N_0 - N_{abs}^{(IR)} \approx 3 \times 10^8$ photons/pulse) is closer to N_0 compared to the Xenon case (where $N_0 - N_{abs}^{(IR)} \approx 2.1 \times 10^8$ photons/pulse), while as expected for HOH generation in Argon the number of "IR-harmonic" peaks was increased and the cut-off position moved to higher harmonics."

Comment #2) I wonder if some of this confusion is caused by mixing up <<detected>> photon numbers, which seem to be denoted on the plots by lower case n , with the <<derived>> photon numbers that appear in the text as a capital N . An example of a derived photon number (at least, according to the given definition), is $N_q^{(IR)}$, defined on line 91 as the number of IR photons absorbed towards harmonic generation. However, to make sense of Fig SI.1. one has to treat $N_q^{(IR)}$ as a detected photon number, read straight off the vertical axis. To put it very plainly: does $N_q^{(IR)}$ increase with q , as the equations on lines 115 and 116 imply, or does it decrease with q , as Fig SI.1. shows?

The y-axis (in Fig.SI.1 of the Suppl. Info) noted with " $n/10^8$ " shows the measured photon number. The black squares correspond to measured photon number n of the peaks of the distribution. $N_q^{(IR)}$ is the absorbed photon number which increases with q . This is now added in each figure caption of the main text and the supplementary info, and furthermore, we augmented figure SI.1 to make it perfectly clear.

Comment #3) Due to the aforementioned issues, I am still not completely certain of the procedure used to process the data and obtain $N_q^{(IR)}$ (for nonzero q and $q=0$), and hence R . However, it seems to me that in performing the linear extrapolation to obtain $N_{(q=0)}^{(IR)}$ (Fig. SI.1.), the authors have implicitly assumed that $R=2$, so that this fact does not serve as a 'check' to validate the spectrum. To put it in more detail: as I understand it, authors measure a distribution of detected IR photon numbers $f(n)$ which contains regularly spaced peaks. Based on comparison to the conventional harmonic spectrum, they assign to the peaks consecutive odd integers q representing harmonic orders. They fit a straight line vs q to the position of the peaks and extrapolate to $q=0$ to obtain $n_{(q=0)}$. They define $N_q^{(IR)}=n_{(q=0)}-n_q$, the

difference in position between a peak and the extrapolated $q=0$ value. If the linear extrapolation is good, then it follows (e.g. from similar triangles) that $\Delta N_q(\text{IR})/2 = N_q(\text{IR})/q$ and hence R (as defined on line 129) equals 2 - regardless of the actual experimental data!

As pointed in our reply to Point #1 of Referee #2, the discussion about R was meant to show that using the calibration procedure described in the manuscript the measured spacing between the consecutive "IR-harmonic" peaks is constant and in agreement with the expected value of consecutive harmonics. In other words, R describes the quality of the linear dependence of N_q versus q , i.e. the fact that the spacing between consecutive IR harmonic peaks is constant. This is not a result of any cyclic argument, but an independent feature of the data.

Nevertheless, we infer from the comments of both referees that the discussion of R is more confusing rather than illuminating, so we opt to remove it and just mention explicitly the constant spacing between consecutive peaks.

Comment #4) Line 119: Does $A(\Delta q)$ mean $A \cdot \Delta q$ or A evaluated at Δq ? If the latter, what does this mean?

It is $A \cdot \Delta q$, we added () in the formula.*

REVIEWERS' COMMENTS:

Reviewer #2 (Remarks to the Author):

The Authors have addressed my concerns. For this reason I recommend the paper for publication in Nature Communication.

Reviewer #3 (Remarks to the Author):

The authors have addressed my comments satisfactorily. I now understand their reasoning and recommend the article for publication.